# Highly pathogenic avian influenza A (H5N1) in marine mammals and seabirds in Peru

Mariana Leguia [1,2] ✉, Alejandra Garcia-Glaessner [1,2],
Breno Muñoz-Saavedra[1,2], Diana Juarez[1,2], Patricia Barrera [1,2],
Carlos Calvo-Mac [2], Javier Jara[3], Walter Silva[3], Karl Ploog[3], Lady Amaro [3],
Paulo Colchao-Claux [4], Christine K. Johnson [2,5], Marcela M. Uhart [2,5],
Martha I. Nelson [6] & Jesus Lescano [3]

Highly pathogenic avian influenza (HPAI) A/H5N1 viruses (lineage 2.3.4.4b) are rapidly invading the Americas, threatening wildlife, poultry, and potentially evolving into the next global pandemic. In November 2022 HPAI arrived in Peru, triggering massive pelican and sea lion die-offs. We report genomic characterization of HPAI/H5N1 in five species of marine mammals and seabirds (dolphins, sea lions, sanderlings, pelicans and cormorants). Peruvian viruses belong to lineage 2.3.4.4b, but they are 4:4 reassortants where 4 genomic segments (PA, HA, NA and MP) position within the Eurasian lineage that initially entered North America from Eurasia, while the other 4 genomic segments (PB2, PB1, NP and NS) position within the American lineage (clade C) that circulated in North America. These viruses are rapidly accruing mutations, including mutations of concern, that warrant further examination and highlight an urgent need for active local surveillance to manage outbreaks and limit spillover into other species, including humans.

The recent emergence of highly pathogenic avian influenza (HPAI) H5N1 viruses in mammals and birds in the Americas[1,2] presents a severe threat to wild and endangered species, to poultry production[3,4], and to public health when the virus spills over into humans[3,5]. The H5N1 (clade 2.3.4.4b) virus arrived in North America in late 2021 from Eurasia and spread across the continent in wild birds, spilling over into poultry farms and infecting an alarming number of wild terrestrial mammals, including fox, skunk, bear, bobcat, and raccoon[6–8]. In October 2022, an apparent mink-to-mink transmission of clade 2.3.4.4b H5N1 viruses in Spain[9] further heightened concern that the avian virus was adapting to mammals and that an H5N1 global pandemic in humans could be approaching.

In November 2022, Peruvian pelicans (*Pelecanus thagus*) along the coast and in offshore islands began experiencing a mass die-off[10,11].

HPAI A/H5N1 was identified in samples collected from dead birds, resulting in the country declaring a sanitary alert[10]. This was followed by several spillover events into other domestic and wild birds, including zoo animals and wild raptors[4,11]. By the beginning of 2023, the Peru outbreak had spread to marine mammals, particularly affecting the South American sea lion (*Otaria flavescens*), which also began to experience a mass die-off[12].

The Pacific coast of Peru hosts a rich biodiversity of marine mammals and seabirds[13]. The ecosystem is home to large populations of South American sea lion, "guano birds" like the Guanay cormorant (*Phalacrocorax bougainvillii*), the Peruvian booby (*Sula variegata*) and the Peruvian pelican, and endangered birds like the Humboldt penguin (*Spheniscus humboldti*). The region also serves as stopover points and feeding grounds for diverse avian migratory species, including the

[1]Laboratorio de Genómica, Pontificia Universidad Católica del Perú (PUCP), Lima, Peru. [2]EpiCenter for Emerging Infectious Disease Intelligence, Centers for Research in Emerging Infectious Diseases, Lima, Peru. [3]Servicio Nacional Forestal y de Fauna Silvestre (SERFOR), Ministerio de Desarrollo Agrario y Riego (MIDAGRI) del Perú, Lima, Peru. [4]Wildlife Conservation Society (WCS) – Perú, Lima, Peru. [5]One Health Institute, School of Veterinary Medicine, University of California, Davis, CA, USA. [6]National Center for Biotechnology Information, National Library of Medicine, National Institutes of Health (NIH), Bethesda, MD, USA. ✉e-mail: mariana.leguia@pucp.edu.pe

Franklin's gull (*Leucophaeus pipixcan*) and several species of sandpipers (*Calidris* spp.)[14,15] that travel south from the northern hemisphere during the boreal winter. Endemic avian influenza A viruses circulate in birds in Peru[16], including unusual reassortant viruses with combinations of genes from the dominant "American" lineage that is widespread in North America and the "South American" lineage that is commonly found in waterfowl in Argentina and Chile[17,18]. However, HPAI viruses of Eurasian origin that caused a major US outbreak in 2014–2015[19] never reached Peru and, to our knowledge, the 2022–2023 HPAI outbreak in South America represents the first incursion of HPAI in the region.

Here we report detection, genomic characterisation, phylogenetic analysis, and mutation analysis of HPAI A/H5N1 viruses identified in marine mammals (sea lion and common dolphin) and seabirds (pelican, cormorant and sanderling) sampled along the coast of Peru since November 2022.

## Results

### Detection of HPAI positive samples in mammals and seabirds in Peru

In response to a multi-species outbreak starting in November 2022, we collected 69 swabs of external orifices and internal organ tissues from seven species of marine mammals and seabirds (common dolphin (*Delphinus delphis*), South American sea lion *(Otaria flavescens)*, sanderling *(Calidris alba)*, Peruvian pelican (*Pelecanus thagus*), Guanay cormorant (*Phalacrocorax bougainvillii*), Peruvian gull (*Larus belcheri*) and Humboldt penguin (*Spheniscus humboldti*)) in regions representing the northern (Piura), central (Lima) and southern (Arequipa and Tacna) coast of Peru (Supplementary Table 1, Fig. 1A). All animals sampled were either deceased or manifested clear signs of disease, including respiratory, digestive, and/or neurological symptoms indicative of acute encephalitis. Seabirds exhibited disorientation, ataxia, circling, nystagmus, torticollis, congested conjunctivae and dyspnea; sea lions exhibited disorientation, ataxia, circling, "stargazing" posture, copious nasal discharge, sialorrhea, dyspnea and seizures; the dolphin was found recently deceased.

To date, we have tested 69 samples from a total of 28 individuals by RT-qPCR, confirming 11 individuals (1 dolphin, 4 sea lions and 6 seabirds), plus 1 pooled sample from 5 sea lions, as positive for influenza A. All samples were also tested for coronaviruses, alphaviruses, bunyaviruses and flaviviruses using pan-PCR assays, and all samples were negative for these tests. Influenza A positives were subjected to NGS for subtyping and to generate full genomes for phylogenetic and mutational analysis. We confirmed 11 individuals as positive for HPAI A/H5N1 (1/1 dolphin, 3/4 individual sea lions, 1/1 pooled sample from 5 sea lions and 6/6 seabirds). It was not possible to type the remaining positive sea lion, as the sample contained very low viral loads or extensive signs of nucleic acid degradation (Supplementary Table 1), likely associated with extensively decomposed tissues in deceased animals (Fig. 1B). Of the 11 individuals that yielded quality sequence data, we generated complete sequences for most genomic regions in most samples (Table 1 and Supplementary Table 1). All sequences have been deposited in GenBank (Accession Numbers OQ550419-OQ550478, and OQ925704-OQ925729).

### Eurasian-American lineage reassortants

To further characterise our Peruvian HPAI A/H5N1 isolates we carried out phylogenetic analyses for each genome segment separately, using all HPAI A/H5N1 reference sequences available globally for avian and mammalian H5 viruses submitted to GISAID since January 1, 2021, including the original A/H5N1 goose/Guangdong strain identified in 1996[20]. Our analysis confirms that all samples sequenced are closely related and cluster within the HPAI A/H5N1 lineage 2.3.4.4b, and further, that the eight Peruvian HPAI A/H5N1 viruses are reassortants (Figs. 2 and 3). Peruvian viruses are positioned in the Eurasian lineage in trees inferred for the PA, HA, NA and MP segments, and cluster within the large clade of North American viruses that were first introduced from Eurasia to the North American's Atlantic flyway in late 2021 (Fig. 2). In contrast, Peruvian viruses are positioned in the American lineage for trees inferred for the PB2, PB1, NP and NS segments, which is evidence of a 4:4 reassortment event. Most of the H5N1 viruses observed since the summer of 2022 in the Americas are reassortants

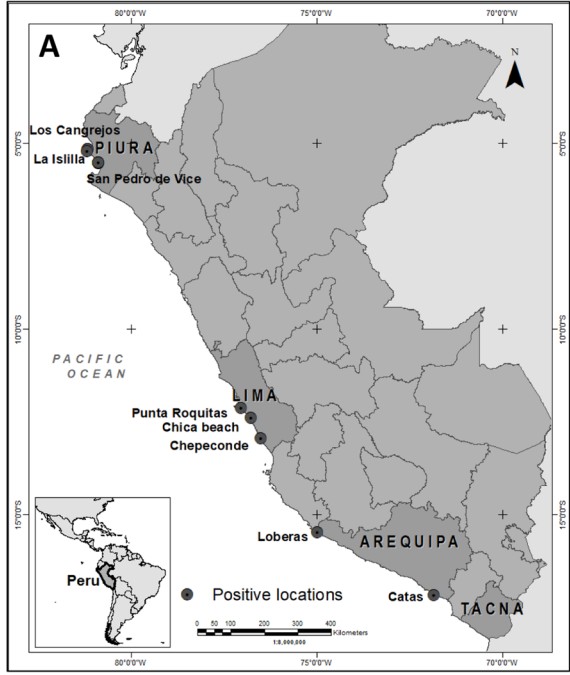

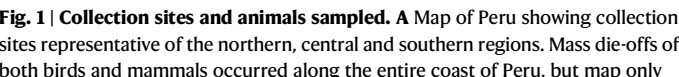

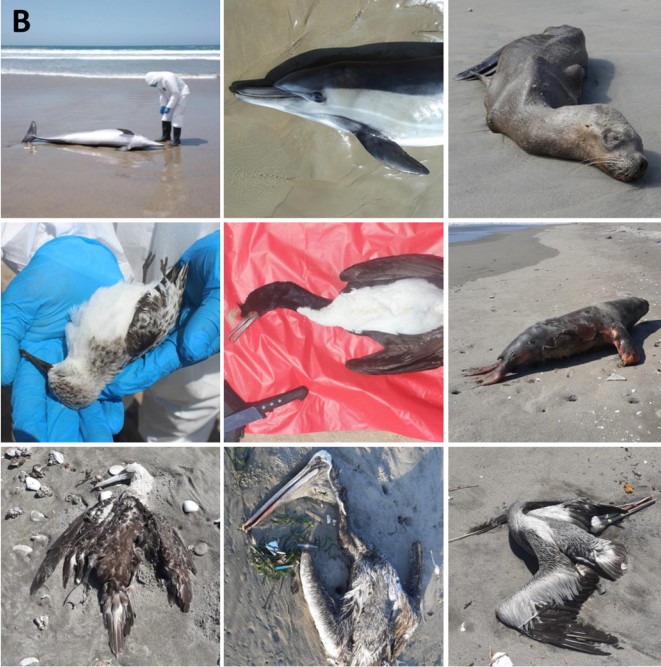

**Fig. 1 | Collection sites and animals sampled. A** Map of Peru showing collection sites representative of the northern, central and southern regions. Mass die-offs of both birds and mammals occurred along the entire coast of Peru, but map only shows positive locations of animals samples in this study. **B** Photographic record of animals sampled, including common dolphin, South American sea lion, sanderling, Guanay cormorant, Peruvian booby and Peruvian pelican.

**Table 1 | Eleven Peruvian HPAI A/H5N1 genomes sequenced in this study**

| Virus name | Species | Sample | Found | Sex | Age | Date | Location | Accession |
|---|---|---|---|---|---|---|---|---|
| A/common dolphin/Peru/PIU-SERO02/2022 | D.delphis | Rectal | Dead | M | Adult | 2022/11/22 | Piura (north) | OQ550439-OQ550446 |
| A/sanderling/Peru/PIU-SERO05/2022 | C. alba | Trachea | Dead | n/a | Juvenile | 2022/11/22 | Piura (north) | OQ550463-OQ550470 |
| A/pelican/Peru/PIU-SERO13/2022 | P. thagus | Brain | Dead | M | Adult | 2022/11/23 | Piura (north) | OQ550419-OQ550422, OQ925704-OQ925707 |
| A/guanay cormorant/Peru/PIU-SERO24/2022 | P. bougainvillii | Brain | Dead | F | Adult | 2022/11/24 | Piura (north) | OQ550423-OQ550430 |
| A/pelican/Peru/PIU-SERO19/2022 | P. thagus | Brain | Dead | M | Adult | 2022/11/24 | Piura (north) | OQ550447-OQ550454 |
| A/pelican/Peru/PIU-SERO28/2022 | P. thagus | Brain | Dead | F | Adult | 2022/11/24 | Piura (north) | OQ550455-OQ550462 |
| A/pelican/Peru/PIU-SERO16/2022 | P. thagus | Lung | Dead | F | Adult | 2022/11/24 | Piura (north) | OQ550431-OQ550438 |
| A/south american sea lion/Peru/LIM-SERO36/2023 | O. flavescens | Lung | Abortion (dead) | M | Abortion | 2023/01/23 | Lima (central) | OQ550471-OQ550478 |
| A/south american sea lion/Peru/LIM-SERO0B/2023 | O. flavescens | Oral - Rectal | Alive | M | Sub-Adult | 2023/01/25 | Lima (central) | OQ925716-OQ925721 |
| A/south american sea lion/Peru/AQP-SERO0K/2023 | O. flavescens | Nasal Pool | 2 Dead/3 Alive | n/a | 2 Adult, 2 Pups, 1 Juveline | 2023/02/07 | Arequipa (south) | OQ925722-OQ925729 |
| A/south american sea lion/Peru/AQP-SERO0R/2023 | O. flavescens | Nasal | Dead | M | Adult | 2023/03/06 | Arequipa (south) | OQ925708-OQ925715 |

Supplementary Table 4 contains all GenBank accession numbers individually listed with hyperlinks.

containing various combinations of segments from the Eurasian and American lineages (Fig. 3). The Eurasian lineage H5N1 virus that originally invaded North America in 2021 has reassorted multiple times with the endemic American lineage since arriving in North America, as evidenced by multiple reassortant clades positioned within the American lineage (Fig. 2). The Peru viruses are positioned in clade C on the PB2, PB1, NP and NS trees and their 4:4 reassortment pattern is referred to as "R6" (Fig. 3).

R6 reassortants were first detected in North American poultry in March 2022 (e.g., EPI_ISL_11971482|A/turkey/South Dakota/22-008485-002/2022) (Fig. 4A), spread regionally across the central United States during the spring and summer of 2022 (Fig. 4B), and became the dominant genotype among sequenced H5N1 viruses in the Americas during the autumn of 2022 (Fig. 4C). The population genetics of H5N1 in the Americas underwent a shift during 2022, as reassortants displaced the original non-reassortant Eurasian H5N1 virus, which has not been detected in the Western hemisphere since June 2022. All currently detected H5N1 reassortants in the Western hemisphere still retain Eurasian HA, NA, and MP segments, but their PB2 and NP segments belong to the American lineage. Both Eurasian and American lineages continue to co-circulate in the Americas for the PB1, PA, and NS segments in various reassortant backgrounds (Fig. 3). Viruses isolated from poultry in Colombia and Ecuador[21] in November 2022 (e.g., A/chicken/Ecuador/02/2022) have the same R6 reassortant genotype as the Peru viruses that were also first detected in November 2022. However, the South America H5N1 viruses are not monophyletic (Fig. 4A) and represent at least four independent viral introductions from North American birds migrating to South American countries during fall migration (1 introduction to Ecuador, 1 introduction to Chile/Peru (see below), and 2 introductions to Colombia). The H5N1 viruses isolated from pelicans in Venezuela in November 2022 have a different reassortant genotype (R7) that is primarily found in the eastern US, including in Florida, and represents a fifth independent H5N1introduction from North to South America during the fall of 2022 (Fig. 3).

H5N1 viruses from Peru and Chile with the R6 genotype form a single clade, representing the only phylogenetic evidence of H5N1 spread between two South American countries to date (Fig. 4A). The Peru-Chile clade appears to descend from a single viral introduction from North America, estimated to have occurred during October 2022 (95% HPD, September 05, 2022 to October 23, 2022). Whether the introduction from North America first arrived in Peru or Chile is difficult to infer from the tree, since the posterior probabilities are similar for both locations at this node (0.41 for Peru; 0.47 for Chile). Within the Peru-Chile clade, the H5N1 viruses collected from northern Peru (Piura region) and central Peru (Lima region) cluster together, consistent with transmission in Peru between avian and mammalian species (pelican, cormorant, sandpiper, dolphin and sea lion). However, two H5N1 viruses collected from sea lions in southern Peru (Arequipa region) during the latter phases of the outbreak cluster with Chilean viruses, reflecting instances of viral gene flow across Peru's southern border. Both of these viruses have the PB2 D701N mutation that is associated with enhanced mammalian transmission in mammals (see below), however, these Arequipa viruses were collected in sea lions a month apart (February 7, 2023 vs. March 6, 2023) and they do not cluster together on the same phylogenetic trees. One of the Arequipa viruses (A/South American sea lion/PeruAQP-SERO0R/2023) with the D701N mutation clusters on the PB2 tree with the H5N1 virus isolated from a human in Chile[22] a few weeks later (March 24, 2023) that also has the D701N mutation (Figs. 4A and 5), but this pattern does not hold for trees inferred using other genome segments. Taken together, these data provide evidence that the D701N mutation is emerging repeatedly in H5N1 viruses that infect mammalian hosts along South America's Pacific coast, however, these data cannot yet ascertain whether viruses with this mutation are transmitting within mammals.

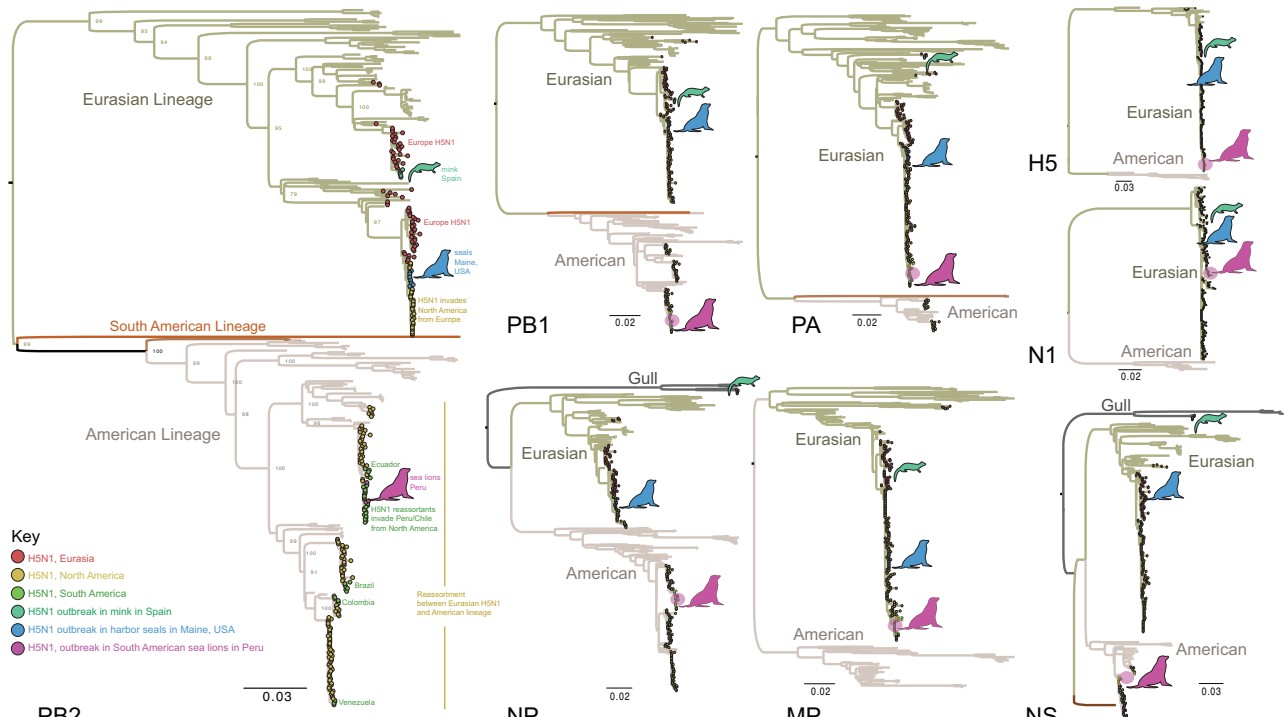

**Fig. 2 | Global H5N1 phylogenies for all eight genome segments.** Phylogenetic trees for AIVs collected globally and submitted to GISAID since January 1, 2021, inferred using ML methods. Branches shaded by AIV lineage; tips shaded by AIV subtype and location. H5N1 viruses from the 2021–2023 Western hemisphere outbreak are shaded yellow, South American H5N1 viruses (except Peru) are shaded green, and Peruvian H5N1 viruses are shaded pink. Cartoon animals denote notable outbreaks in mink in Spain, sea lions in Peru, and seals in Maine, USA. A detailed tree is provided for the PB2 segment, including bootstrap values, labels, and a box around the R6 reassortant clade that is presented in greater detail in Fig. 3A. Smaller trees are provided for the other 7 segments. Raw tree files for each genome segment are available at GitHub (https://github.com/mostmarmot/Peru_AIV/).

## Mutation analysis

We also performed a detailed SNP and mutational analysis to identify amino acid changes potentially linked to increased virulence, transmission, or mammalian host adaptation, and to assess if we could identify specific differences between host species (mammals vs. birds). We focused on variable sites that were different from both the original A/H5N1 goose/Guangdong strain from 1996[20] and the A/Vietnam/1203/2004 strain used to annotate amino acid positions in the CDC inventory[23]. Based on these criteria, we identified more than 70 variable sites spread across all genomic regions (Supplementary Table 2), including 21 that have been previously reported as linked to specific phenotypes, such as altered polymerase activity and replication efficiency (usually enhanced), increased virus binding to α2–3 and α2–6, enhanced transmission, and increased virulence and pathogenicity, including in mammals[23,24]. However, given the increased age of the two reference sequences used to define variable sites, which would make them unsuitable to identify potentially relevant recent changes, we also included in the analysis two additional references representing R6 reassortants from 2022 identified in birds (A/chicken/Wyoming/22-032216-001/2022) and mammals (A/skunk/Washington/22-019274-001/2022). Based on this more stringent comparison, we identified 23 variable positions in Peruvian viruses that contain mutations that are not present in either the Guangdong/Vietnam reference strains from 1996/2004, or the R6 reassortants from 2022 (Fig. 5, Supplementary Table 2). Within this group, 15 mutations are present in all birds and mammals from Peru (3 cases) or present in single individuals from Peru (12 cases). The remaining 8 sites are of particularly high interest because they repeat two or more times. These sites include 1 in PB2 (D701N), 1 in PB1 (S515A), 3 in PA (R57Q, T85V, M86I), 1 in HA (H355R, according to the H5 numbering scheme), 1 in NP (Y289F) and 1 in NA (A81I), and half of them cluster solely in mammalian viruses from Peru.

Particularly noteworthy is the PB2 D701N mutation, which appears in sea lion samples collected during the late phases of the outbreak in Peru in February and March, and subsequently appears in the virus sequenced from the human case reported in Chile[22]. This mutation, along with two other PB2 mutations (E627K, K702R) not present in our data, have been specifically linked to mammalian host adaptation and enhanced transmission previously[24,25]. In addition, despite not finding PB2 E627K and K702R, we have identified two additional mutations (PA M861I and NS1 D26K) in mammalian samples from Peru that are also present in the human case from Chile[22], raising the possibility that the viruses are changing in a host-specific manner[24,25] that supports mammalian host adaptation. The sea lion viruses contain additional unique mutations, such as PB1 L378M and S515A, that may warrant further observation, as these are also showing up in Chilean genomes, mostly from birds. Sampling of additional individuals will be needed to fully assess the significance of these uncharacterised variable sites, especially in terms of their pathogenicity to mammals. However, it is clear that most are restricted to sequences reported in Latin America (Fig. 5, Supplementary Table 2), which indicates that the virus is indeed changing, and perhaps even adapting to mammals, as it travels south from the northern hemisphere.

## Discussion

For decades, South America's relative geographic isolation has largely insulated its fragile coastal ecosystems and poultry industry from the HPAI outbreaks that periodically ravage US and Mexican farms to the north. But the newest variant of HPAI A/H5N1, clade 2.3.4.4b, is spreading faster, causing mass mortality in wildlife, infecting mammals, and invading countries like Peru that had remained HPAI-free for decades. The arrival of HPAI in regions with less experience managing highly pathogenic viruses in wildlife and poultry is highly concerning.

| | Genotype | First Detection | Notable viruses | Flyways | Countries |
|---|---|---|---|---|---|
| **Eurasian** n=245 | [1 2 3 4 / 5 6 7 8] (all grey) | 17-Dec-2021 | A/emu/NL/FAV-0035-12/2021<br>A/harbor seal/Maine/22-020455-003/2022<br>A/grey seal/Maine/22-020983-003/2022 | Atlantic, Mississippi, Central, Pacific | Canada<br>United States |
| **R1** n=37 | [B B _ _ / B _ _ _] | 12-Feb-2022 | A/fox/New York/074441/2022 | Atlantic, Mississippi, Central | United States |
| **R2** n=54 | [B B B _ / B _ _ _] | 01-Mar-2022 | A/bottlenose_dolphin/Florida/UFTt2203/2022<br>A/Virginia opossum/Iowa/22-016780-001/2022<br>A/fox/Iowa/22-015357-002/2022<br>A/fox/Michigan/22-014536-001/2022<br>A/fox/New York/115912/2022 | Atlantic, Mississippi, Central | United States |
| **R3** n=250 | [A _ _ _ / A _ _ _] | 04-Mar-2022 | A/fox/Minnesota/22-014660-001/2022<br>A/red fox/MB/FAV-0414-12/2022<br>A/raccoon/Washington/ | Mississippi, Central, Pacific | Canada<br>United States |
| **R4** n=53 | [D _ _ _ / D _ _ _] | 05-Mar-2022 | A/Skunk/MB/FAV-470/2022 | Central, Pacific | Canada<br>United States |
| **R5** n=39 | [C _ _ _ / C _ _ _] | 12-Mar-2022 | A/red fox/North Dakota/22-017354-001/2022 | Mississippi, Central Pacific | United States |
| **R6** n=171 | [C C _ _ / C _ _ C] | 21-Mar-2022 | **A/bottle-nose dolphin/Peru/PIU-SER002/2022**<br>**A/South American sea lion/Peru/LIM-SER036/2023**<br>A/Peru/PIU-001/2022<br>A/skunk/Washington/22-019274-001/2022<br>A/striped skunk/Idaho/22-037165-002/2022<br>A/bobcat/Wisconsin/22-016051-001/2022<br>A/red fox/MB/FAV-370-01/2022<br>A/fox/Minnesota/22-014182-001/2022<br>A/red fox/Michigan/22-018712-001/2022 | Atlantic, Mississippi, Central, Pacific | Ecuador<br>Canada<br>Peru<br>United States |
| **R7** n=27 | [A _ _ _ / A _ _ A] | 07-Apr-2022 | A/Pelican/Venezuela/Pel3/2022 | Atlantic, Mississippi, Central | United States<br>Venezuela |
| **R8** n=42 | [A A A _ / B _ _ _] | 27-Sep-2022 | A/skunk/Minnesota/22-034925-001/2022<br>A/skunk/Wisconsin/22-037029-001/2022 | Atlantic, Mississippi, Central | United States |

**Fig. 3 | H5N1 genotypes identified in the Americas, 2021–2023.** Each genotype box represents one of eight genome segments (1–8), shaded by lineage: grey = Eurasian lineage; pink = American lineage (clade A); blue = American lineage (clade B); green = American lineage (clade C); orange = American lineage (clade D).

These events should prompt immediate cross-sectoral capacity strengthening and coordinated response activities throughout the region. Here, we rapidly established new surveillance partnerships between government and academia to respond to mass mortality events involving Peruvian pelicans and South American sea lions. We confirmed the presence of HPAI A/H5N1 clade 2.3.4.4b in both pelicans and sea lions, as well as in Guanay cormorants, sanderlings and dolphins, and further surmised that HPAI A/H5N1 is the likely causative agent of the mass wildlife die-offs. We suspect that direct HPAI transmission between sea lions could be occurring, rather than independent spillovers into sea lions from avian sources, but additional sequence data and analysis will be required to further characterise mammal-to-mammal transmission. We report more than 70 variable sites, many of which have been previously linked to altered polymerase activity and replication efficiency, increased virus binding to α2–3 and α2–6, enhanced transmission, and increased virulence and pathogenicity, including in mammals[23,24,26], but we specifically focus on 23 sites that are different even from R6 resassortants that circulated in 2022. Within this subset we are particularly concerned by the presence of PB2 D701N in 2 sea lion samples, and in a human case reported in Chile[22], as this mutation has been specifically linked to mammalian host adaptation and enhanced transmission[24,25]. The same concern applies to PA M861I and NS1 D26K, which are present in mammalian samples from Peru and in the human case from Chile[22]. Two other mutations of concern (PB2 E627K and K702R), linked to mammalian host adaptation and enhanced transmission[24,25], as well as the PB2 T271A mutation observed in the mink outbreak in Spain[9], were not present in our samples. Our analysis has focused on recent non-synonymous mutations, however, other mutations, including synonymous substitutions should be further examined, as these can change viral RNA structure and splicing, which in turn can potentially impact viral pathogenesis and fitness.

Our phylogenetic analysis supports a single introduction of 2.3.4.4b into Peru from North America, presumably through the movements of migratory wild birds that travel south during the boreal winter, setting the stage for infection of local sea birds that share habitats with marine mammals. There are multiple possible transmission routes for local transmission among species that involve direct contact or indirect environmental transmission. For one, seabirds share feeding spaces with both sea lions and dolphins, providing ample opportunities for direct contact between animals at sea[27–30]. Direct contact also occurs on islands, islets, and guano headlands, especially in protected areas where large and dense breeding colonies of sea lions and seabirds cohabitate, and where indirect transmission is also possible on land and via guano runoff into the surrounding waters[27,28,30,31]. Another scenario for transmission involves carnivory and scavenging of infected animal carcasses by marine and terrestrial carnivores, as well as by raptors, gulls, and other scavenger birds[2,32,33]. Fishing docks, where fishermen often dispose of waste by dumping it at sea, attract seabirds, sea lions, marine otters and others that come to feed. Many docks along the Peruvian coast also function as tourist attractions, where seabirds and sea lions are purposely fed to create photo opportunities, building large congregations of wild animals that also increase opportunities for contact with humans. The confirmed presence of HPAI A/H5N1 in 2 species of resident guano seabirds, the Peruvian pelican and Guanay cormorant, provides another potential route for future transmission to humans[34,35], as guano is widely used to fertilise crops. Finally, the Peruvian desert coast is home to large poultry operations with millions of chickens adjacent to infected wild and migratory birds, placing this production industry at risk and furthering the potential for animal and human contact with circulating HPAI lineages in the region.

There are outstanding questions about which migratory bird species are involved in the long-distance dissemination of HPAI from

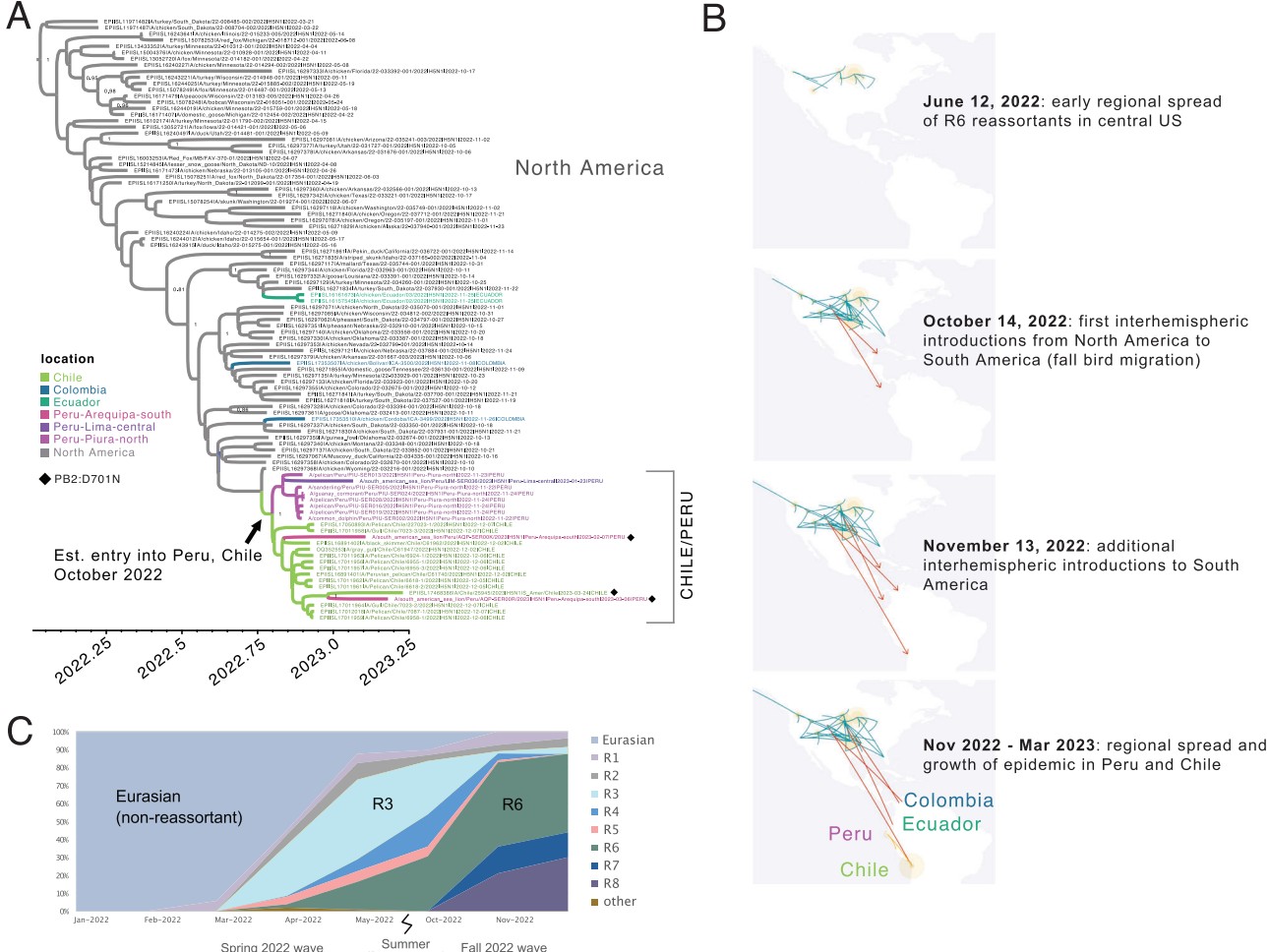

**Fig. 4 | Spread of reassortant "R6" viruses. A** Time-scaled maximum clade credibility (MCC) tree inferred for the PB2 segment using Bayesian approaches for 98 representative H5N1 viruses (R6 genotype) belonging to the reassortant clade C in the American lineage (see Fig. 2). Branches shaded by location. South American virus names shaded by country or Peruvian region. Posterior probabilities provided for key nodes. Estimated timing of R6 virus entry into Peru/Chile is provided.

Diamonds indicate the 3 viruses with PB2:D701N mutations. Raw MCC tree available on GitHub (https://github.com/mostmarmot/Peru_AIV/). **B** Discrete phylogeo-graphic transition history of R6 viruses at four time points. Red lines with arrows indicate interhemispheric transitions from North America to South America. **C** Proportion of H5N1 viruses sequenced in the Americas between 2021–2023 that belong to different genotypes (Fig. 2) over time.

North to South America, possibly by way of Central America. We detected clade 2.3.4.4b in a migratory sanderling (*Calidris alba*) that would have arrived in Peru after breeding in the Canadian arctic. However, *Calidris* spp. are an unlikely conduit for HPAIV A/H5N1 because experimental inoculations result in death or disease within 5 to 11 days of inoculation[36,37]. Given the unlikelihood of a successful long-distance migration for a clinically infected bird, we suspect the sanderling was infected locally. Our phylogenetic analysis supports multiple independent introductions of HPAI from North America into South American countries for which sequence data was available at the time of this study, including Peru, Ecuador, and Venezuela. This contrasts with the single introduction of HPAI from Eurasia to North America observed earlier in 2021. Although North America is the primary source of HPAI for South America's initial HPAI outbreaks, South American countries are likely to become more important sources for regional HPAI outbreaks as the virus establishes locally. After observing HPAI A/H5N1 reassort repeatedly within North American viruses, it is possible that the virus will continue to evolve in South America by mutation and reassortment with the genetically distinct South American AIV lineage that is commonly detected in Argentina[17] and Chile[18]. For this reason, and given the unchar-acterised variable sites reported here, there is an urgent need to

establish pipelines for efficient real-time genomic sequencing of HPAI to track viral evolution and spread across Peru and other countries in South America, as well as funding to support char-acterisation of possible new mutations.

The impact of HPAI A/H5N1 on the morbidity and mortality of endangered species like the Andean condor (*Vultur gryphus*)[38], the Humboldt penguin *(Spheniscus humboldti)*[39], the marine otter (*Lontra felina*)[40], and many others is concerning. The Peruvian coast is one of the few places in the world where scavenger condors feed on dead marine animals[41,42], putting them at risk of infection if they consume contaminated carcasses. The recent infection and death of several critically endangered California condors (*Gymnogyps californianus*) in the United States highlights these risks[43]. The Humboldt penguin lives in large colonies and shares space with guano birds and sea lions, and although the single individual tested here was not positive for HPAI A/H5N1, Chile is observing record mortality in this species, now reaching 10% of their population, with four confirmed positive to HPAI A/H5N1[44]. Similarly, the marine otter[40] is an aquatic mustelid that inhabits the same rocky shores of Peru and Chile inhabited by sea lions. HPAI A/H5N1 has been confirmed previously in otters[2] and recently in two animals from Chile[44], and although none were tested here, we have anecdotal knowledge of otter carcasses washing

| # | | 1 | 2 | 3 | 4 | 5 | 6 | 7 | 8 | 9 | 10 | 11 | 12 | 13 | 14 | 15 | 16 | 17 | 18 | 19 | 20 | 21 | 22 | 23 |
|---|---|---|---|---|---|---|---|---|---|---|----|----|----|----|----|----|----|----|----|----|----|----|----|----|
| | | PB2 | | | | PB1 | | PA | | | HA | | NP | | | | NA | | | M1 | NS1 | | | |
| | | T215M | Q591K* | D701N* | N715T | L378M | S515A | R57Q | T85V | M86I | H355R | A496S | M222L | Y289F | A428T | R452K | V62I | A811 | S339P | N87T | D26K | E60V | I81T | I129M |
| 1 | A/Chile/25945/2023 | · | K | N | · | M | · | Q | A | I | · | · | · | · | · | · | · | T | P | T | K | A | · | · |
| 2 | A/south american sea lion/Peru/AQP-SER00R/2023 | · | K | N | · | M | A | Q | A | I | X | · | · | F | · | · | X | X | P | T | E | A | X | X |
| 3 | A/south american sea lion/Peru/AQP-SER00K/2023 | · | · | N | · | M | · | Q | · | · | R | · | X | X | X | X | · | T | P | T | E | A | · | · |
| 4 | A/south american sea lion/Peru/LIM-SER00B/2023 | X | X | X | X | X | A | Q | V | · | R | · | · | F | · | · | · | T | P | T | K | V | · | · |
| 5 | A/south american sea lion/Peru/LIM-SER036/2023 | M | · | · | T | M | · | · | A | I | · | S | L | · | T | K | I | T | P | T | E | A | · | · |
| 6 | A/Pelican/Venezuela/Inf9/2022 | · | · | · | · | · | · | Q | A | · | · | · | · | · | · | · | · | T | · | · | E | A | X | T |
| 7 | A/Gull/Chile/7023-2/2022 | · | · | · | · | M | A | Q | A | · | · | · | · | · | · | · | · | T | P | T | K | A | · | · |
| 8 | A/Pelican/Chile/7087-1/2022 | · | · | · | · | M | A | Q | A | · | · | · | · | · | · | · | · | T | P | T | K | A | · | · |
| 9 | A/Pelican/Chile/6924-1/2022 | · | · | · | · | M | · | Q | A | · | · | · | · | · | · | · | · | T | P | T | E | A | · | · |
| 10 | A/Pelican/Chile/6618-1/2022 | · | · | · | · | M | A | Q | A | · | · | · | · | · | · | · | · | T | P | T | K | A | · | · |
| 11 | A/Pelican/Chile/6618-2/2022 | · | · | · | · | M | A | Q | A | · | · | · | · | · | · | · | · | T | P | T | K | A | · | · |
| 12 | A/black_skimmer/Chile/C61962/2022 | · | · | · | · | M | · | Q | A | · | · | · | · | · | · | · | · | T | P | T | E | A | · | · |
| 13 | A/gray_gull/Chile/C61947/2022 | · | · | · | · | M | A | Q | A | · | · | · | · | · | · | · | · | T | P | T | K | A | · | · |
| 14 | A/chicken/Peru/LIM-003/2022 | X | · | · | · | M | · | Q | A | · | · | · | · | · | · | · | · | T | P | T | E | A | · | · |
| 15 | A/chicken/Peru/LAM-002/2022 | X | · | · | · | M | · | · | A | · | · | · | · | · | · | · | · | T | L | T | E | A | · | · |
| 16 | A/chicken/Colombia_Cordoba/ICA-3499/2022 | · | · | · | · | · | · | · | A | · | · | · | · | · | · | · | · | T | · | · | E | A | · | · |
| 17 | A/guanay cormorant/Peru/PIU-SER024/2022 | · | · | · | · | M | · | · | A | · | · | · | · | · | · | · | · | I | P | T | E | A | · | · |
| 18 | A/pelican/Peru/PIU-SER019/2022 | · | · | · | · | M | · | · | A | · | · | · | · | · | · | · | · | T | P | T | E | A | · | · |
| 19 | A/pelican/Peru/PIU-SER028/2022 | · | · | · | · | M | · | · | A | · | · | · | · | · | · | · | · | I | P | T | E | A | · | M |
| 20 | A/pelican/Peru/PIU-SER016/2022 | · | · | · | · | M | · | · | A | · | · | · | · | · | · | · | · | T | P | T | E | A | · | · |
| 21 | A/Pelican/Venezuela/Pel4/2022 | · | · | · | · | X | · | Q | A | · | · | · | · | · | · | · | · | T | · | · | E | A | · | T |
| 22 | A/Pelecanus_occidentalis/Venezuela/4S1/2022 | · | · | · | · | X | · | Q | A | · | · | · | · | · | · | · | · | T | · | · | E | A | · | T |
| 23 | A/Pelican/Venezuela/Pel3/2022 | · | · | · | · | · | · | Q | A | · | · | · | · | · | · | · | · | T | · | · | E | A | · | T |
| 24 | A/Pelecanus_occidentalis/Venezuela/3S1/2022 | · | · | · | · | · | · | Q | A | · | · | · | · | · | · | · | · | T | · | · | E | A | · | T |
| 25 | A/chicken/Ecuador/02/2022 | · | · | · | · | · | · | · | A | · | · | · | X | X | X | X | · | T | · | · | E | A | · | · |
| 26 | A/pelican/Peru/PIU-SER013/2022 | · | · | · | · | M | · | · | A | · | · | · | · | H | · | · | · | T | P | T | E | A | T | · |
| 27 | A/common dolphin/Peru/PIU-SER002/2022 | · | · | · | · | M | · | · | A | · | · | · | · | · | · | · | · | T | P | T | E | A | · | · |
| 28 | A/sanderling/Peru/PIU-SER005/2022 | · | · | · | · | M | · | · | A | · | · | · | · | · | · | · | · | T | P | T | E | A | · | · |
| 29 | A/chicken/Colombia_Magdalena/ICA-3503/2022 | · | · | · | · | · | · | · | A | · | · | · | · | · | · | · | · | T | · | · | E | A | · | · |
| 30 | A/chicken/Colombia_Choco/ICA-3502/2022 | · | · | · | · | · | · | · | · | · | · | · | · | · | · | · | · | T | · | · | E | A | · | · |
| 31 | A/pelican/Peru/PIU-001/2022 | X | · | · | · | X | · | · | A | · | · | · | · | · | · | · | · | T | P | T | E | A | · | · |
| 32 | A/chicken/Colombia_Bolivar/ICA-3500/2022 | · | · | · | · | · | · | · | · | · | · | · | · | · | · | · | · | T | · | · | E | A | · | · |
| 33 | A/chicken/Colombia_Choco/ICA-3504/2022 | · | · | · | · | · | · | · | · | · | · | · | · | · | · | · | · | T | · | · | E | A | · | · |
| 34 | A/duck/Colombia_Choco/ICA-3501/2022 | · | · | · | · | · | · | · | · | · | · | · | · | · | · | · | · | T | · | · | E | A | · | · |
| 35 | A/domestic_duck/Florida/22-032962-001/2022 | · | · | · | · | · | · | Q | A | · | · | · | · | · | · | · | · | T | · | · | E | A | · | T |
| 36 | A/chicken/Wyoming/22-032216-001/2022 | · | · | · | · | · | · | · | A | · | · | · | · | · | · | · | · | T | · | · | E | A | · | · |
| 37 | A/skunk/Washington/22-019274-001/2022 | · | · | · | · | · | · | · | A | · | · | · | · | · | · | · | · | T | · | · | E | A | · | · |
| 38 | A/fox/Michigan/22-014536-002-original/2022 | · | · | · | · | · | · | · | · | · | · | · | · | · | · | · | · | T | · | · | E | A | · | · |
| 39 | A/bottlenose_dolphin/Florida/UFTt2203/2022 | · | · | · | · | · | · | · | · | · | · | · | · | · | · | · | · | T | · | · | E | A | · | · |
| 40 | A/black_vulture/Florida/W22-161/2022 | · | · | · | · | · | · | · | A | · | · | · | · | · | · | · | · | T | · | · | E | A | · | · |
| 41 | A/fox/Iowa/22-015357-002-original/2022 | · | · | · | · | · | · | · | · | · | · | · | · | · | · | · | · | T | · | · | E | A | · | · |
| 42 | A/bald_eagle/Kansas/W22-197/2022 | · | · | · | S | · | · | · | A | · | · | · | · | · | · | · | · | T | · | · | E | A | · | · |
| 43 | A/grey_seal/Maine/22-020983-003-original/2022 | · | · | · | S | · | · | · | A | · | · | · | · | · | · | · | · | T | · | · | E | A | · | · |
| 44 | A/harbor_seal/Maine/22-020455-003/2022 | · | · | · | S | · | · | · | A | · | · | · | · | · | · | · | · | T | · | · | E | A | · | · |
| 45 | A/harbor_seal/Maine/22-020455-001/2022 | · | · | · | S | · | · | · | A | · | · | · | · | · | · | · | · | T | · | · | E | A | · | · |
| 46 | A/harbor_seal/Maine/22-020455-005/2022 | · | · | N | S | · | · | · | A | · | · | · | · | · | · | · | · | T | · | · | E | A | · | · |
| 47 | A/red_fox/Netherlands/22002190-003/2022 | · | · | · | · | · | · | · | · | · | · | · | · | · | · | · | · | T | · | · | E | A | · | · |
| 48 | A/red_fox/Netherlands/22003495-003/2022 | · | · | · | · | · | · | · | · | · | · | · | · | · | · | · | · | T | · | · | E | A | · | · |
| 49 | A/otter/Netherlands/22001014-005/2022 | · | · | · | · | · | · | · | · | · | · | · | · | · | · | · | · | T | · | · | E | A | · | · |
| 50 | A/mink/Spain/22VIR12774-14_3869-3/2022 | · | · | · | · | · | · | · | T | · | · | · | · | · | · | · | · | T | · | · | · | · | S | T |
| 51 | A/porpoise_/Sweden/SVA220712SZ0367/O-2022 | · | · | · | · | · | · | · | A | · | · | · | · | · | · | · | · | T | · | · | E | A | · | · |
| 52 | A/Mandarin_duck/Korea/WA496/2022 | · | · | · | · | · | · | · | A | · | · | · | · | · | · | · | · | T | · | · | E | A | · | T |
| 53 | A/red_fox/Netherlands/21040099-001/2021 | · | · | · | · | · | · | · | A | · | · | · | · | · | · | · | · | T | · | · | E | A | · | · |
| 54 | A/Great_white_pelican/Israel/619/2021 | · | · | · | · | · | · | · | A | · | · | · | · | · | · | · | · | T | · | · | X | X | X | X |
| 55 | A/mandarin_duck/Korea/WA585/2021 | · | · | · | · | · | · | · | A | · | · | · | · | · | · | · | · | T | · | · | E | A | · | T |
| 56 | A/Buteo_buteo/belgium/334_0013/2021 | · | · | · | · | · | · | · | A | · | · | S | · | · | · | · | · | T | · | · | E | A | · | · |
| 57 | A/chicken/Bangladesh/49967/2021 | · | · | · | · | · | · | · | · | · | · | · | · | · | · | · | -- | · | · | · | E | T | T | N |
| 58 | A/duck/Laos/NL-2072064/2020 | · | · | · | · | · | · | · | V | · | · | · | · | · | · | K | -- | · | · | · | E | A | -- | · |
| 59 | A/Vietnam/1203/2004 | · | · | · | · | · | · | · | · | · | · | · | · | · | · | · | -- | · | · | · | E | A | -- | · |
| 60 | A/Goose/Guangdong/1/96 | T | Q | D | N | L | S | R | T | M | H | A | M | V | A | R | V | A | S | N | D | E | I | I |

onto Peruvian shores. Fortunately, marine otters do not live in large groups[45], which might limit intraspecies contagion. However, direct mammal to mammal transmission has been suggested as a possible explanation for an outbreak in a Spanish farm among breeding mink, another mustelid, though evidence of mammal-to-mammal transmission was inconclusive[9]. Various cases of HPAI A/H1 in mammals, like the dolphin case reported here and others in Chile[44], along with the massive South American sea lion die-off in Peru supports direct mammal to mammal transmission as a possible viral dissemination route, but confirmation will require further investigations with larger samples sizes, deeper genomic analyses and epidemiologic data that are often difficult to come by in outbreak settings. Peruvian pelicans

**Fig. 5 | SNP and mutational analysis of Peruvian HPAI a/H5N1 viruses (shown in bold).** Additional reference sequences used are available through GenBank and GISAID. A total of 22 variable sites across genomic segments were identified relative to the original A/H5N1 goose/Guangdong reference from 1996, the A/Vietnam/1203/2004 reference used to annotate amino acid positions in the CDC inventory[23], plus two additional more recent references representing R6 reassortants from 2022 identified in birds (chicken/wyoming/2022) and mammals (skunk/washington/22-019274-001/2022) (shaded in grey). Mutation PB2 D701N (shown with an *) has been previously linked to mammalian host adaptation and enhanced transmission[24,25]. The remaining 21/22 sites have not been previously characterised, and 4 may warrant special consideration (shown in bold) as they are concentrated in recent mammalian samples (Supplement Table 2), including 2 (PB2 D701N and PA M86I) that later show up in the genome sequenced from the human case in Chile (A/Chile/25946/2023)[22]. Dots represent amino acids identical to those present in annotation reference A/Goose/Guangdong/1/96; colours have been randomly assigned to amino acids simply to aid in identification of differences; X indicates there is no genomic information available for that position; - - indicates a deletion in the sequence.

have also suffered massive die-offs at the beginning of the outbreak in Peru, later followed by Guanay cormorants and Peruvian boobies. Peruvian pelicans are considered near-threatened worldwide[46], but in Peru they are classified as endangered due to recent large population declines resulting from severe El Niño events associated with global warming and overfishing of their main source of food, the Peruvian anchovy[15]. For these reasons, efforts are urgently needed to assess the impact of ongoing mass mortalities on Peruvian marine sea birds and mammals.

Finally, an even larger concern is the possibility of spillover into human populations, as has been already documented[22,47,48], followed by massive human-to-human transmission. Previous human cases have resulted in fatalities[49], which has led the WHO to declare that the current zoonotic threat from HPAI A/H5N1 remains elevated and that member states should remain vigilant and consider mitigation steps to reduce human exposure[50]. In Peru, the outbreak occurred along the Pacific coast and during the austral summer, when many people go to the beach. It is not uncommon for beachgoers (and their pets) to interact with sick and disoriented animals without any knowledge of the risks, or for free-roaming dogs in rural and semi-rural coastal areas to encounter sick or dead animals as they scavenge for food. This has led government authorities to relocate live animals that show up in places where they do not belong, or to euthanize sick individuals and appropriately dispose of their carcasses. However, both the pelican and sea lion die-offs have been so massive that it has been very challenging for the authorities to respond in a timely manner. The recent confirmed HPAI H5N1 human case in Chile[22] has been considered an environmental exposure due to an abundance of dead wildlife in the area close to the patient's residence, as the human virus sequenced was identical to local wild bird strains. Heightened public awareness campaigns are needed, including educating the public to avoid contact with infected animals[51]. Animal workers, particularly municipal personnel tasked with cleaning duties, need additional training in the proper use of personal protective equipment, and on management and disposal of infected carcasses[51,52]. People in contact with sick and dead animals infected with HPAI A/H5N1 are at risk of infection, and human cases could be missed in the absence of active and obvious human-to-human transmission. Syndromic surveillance based on PCR-based diagnostics and/or serology of suspected cases in people working in wildlife and poultry outbreak response is highly recommended for early detection of zoonotic transmission. Subsequent serologic studies of people in close contact with infected animals, especially in outbreak settings, can also inform on the extent of zoonotic spillover and direct future surveillance at important animal-human interfaces. This approach promotes opportunities to strengthen ties among governmental institutions at all levels, expert researchers within academia, and the larger scientific community operationalizing a One Health approach, which are essential for epidemic and pandemic preparedness. The work reported here exemplifies such collaboration, and although many of the variable genomic sites identified in Peruvian H5N1 viruses will need further validation with both downstream in-vitro and in-vivo studies, this study of HPAI A/H5N1 in marine birds and mammals in Peru represents a significant contribution to the understanding of a highly pathogenic virus with serious pandemic potential for humans.

## Methods

### Sample collection and pre-processing for influenza A

Starting on November 22, 2022, samples were collected opportunistically from both live and dead animals by trained veterinarians from the Peruvian wildlife service (Servicio Nacional Forestal y de Fauna Silvestre, SERFOR) and the NGO Wildlife Conservation Society Perú, under permit 2023-009134, using full personal protective equipment and following standard protocols for cleaning, disinfection, and disposal of hazardous waste[52]. Some severely clinically ill animals were humanely euthanized[53]. Briefly, animals were chemically immobilised by intramuscular administration of ketamine and xylazine, followed by intravenous or intracardiac injection of T-61™ (MSD Animal Health USA)[53]. Samples and tissues were collected into cryovials containing 0.5 mL of DNA/RNA Shield (Zymo R1200 125) or viral transport media (VTM), and transported at 4 °C within 1–10 days of collection to the laboratory for testing. Nucleic acids were extracted using Quick-DNA/RNA Viral Extraction Kits (Zymo D7021) and tested for influenza A by RT-qPCR using CDC protocols[54] on a Bio-Rad CFX96 instrument.

### Influenza A subtyping

Samples positive for influenza A by RT-qPCR were subtyped using a combination of directed amplification with universal primers targeting conserved genomic regions[55–57], followed by next-generation sequencing (NGS). All primer sequences used are provided in Supplementary Table 3. Briefly, RNA samples were reverse transcribed using Superscript IV (Invitrogen 18090050) and amplified using Q5 High-fidelity DNA polymerase (NEB M0491L). Amplification products were prepared into barcoded sequencing libraries using DNA Prep Kits (Illumina 20060059) and Nextera DNA UD Indexes (Illumina 20027215). The resulting libraries were quality controlled using High Sensitivity DNA kits (Agilent 5067–4626) on a Bioanalyzer 2100 instrument. Libraries were normalised to 4 nM each, pooled, re-quantified using Qubit 1x dsDNA HS Kits (Invitrogen Q33230), and sequenced using High Output Sequencing Kits (Illumina FC-420-1003) on an Illumina MiniSeq instrument.

### Bioinformatics processing

Illumina paired-end raw sequence data was pre-processed to trim sequencing adaptors and filter out low quality/low complexity reads (Phred scores <Q20, 75 bp minimum length) using Geneious Prime 2023.0.4 and BBDuk[58]. Pre-processed reads were then filtered by reference-mapping to various HA (H1, H2, H3, H5, H7) and NA (N1, N2, N3, N5, N7) references (Accession Numbers: NC_026433.1, NC_007374.1, NC_007366.1, NC_007362.1, NC_026425.1, AF144304.1, NC_007382.1, OP806485, MF046172.2, OP723829.1), and to single references in the case of all other segments (Accession Numbers MT624412.1, MN254461.1, KY635563.1, MT825070.1, MW110227.1, MT982385.1). Filtered reads were then re-assembled de-novo using SPAdes[59] to generate complete genomes whenever possible, and to further confirm subtyping by BLAST. All sequences have been deposited in GenBank (Accession Numbers OQ550419-OQ550478, and OQ925704-OQ925729).

### Phylogenetic and mutation analysis

HPAI H5N1 reference sequences were obtained from GenBank and GISAID, and together with the sequences generated here, they were

concatenated and aligned using MAFFT[60]. Maximum likelihood (ML) trees were prepared with FastTree[61] incorporating a general time-reversible (GTR) model of nucleotide substitution with gamma-distributed rate variation among sites. Trees were annotated using the WHO/OIE/FAO nomenclature for highly pathogenic avian influenza A H5[62]. To place the Peruvian viruses in a global context, we downloaded (on February 14, 2023) an additional background dataset of influenza A virus genomes that included all sequences from avian and mammalian H5 viruses submitted to GISAID since January 1, 2021. To keep the dataset up to date, additional South American sequences were added on April 21, 2023. Partial sequences were excluded. Phylogenetic relationships were inferred for each of the eight genome segments using the ML methods available in IQ-Tree 2[63] with a GTR model and a gamma distribution as described above. Due to the size of the dataset, we used the high-performance computational capabilities of the Biowulf Linux cluster at the National Institutes of Health (http://biowulf.nih.gov). To assess the robustness of each node, a bootstrap resampling process was performed with 1000 replicates. Finally, mutation analysis was done using the CDC H5N1 genetic changes inventory for SNP analysis and various other previously published mutations of concern[23,24].

### Bayesian analysis

To examine the evolution of the clade of reassortant R6 viruses in Peru and other locations in greater detail, we performed a time-scaled Bayesian analysis on a dataset of 98 PB2 sequences from the R6 clade. The dataset included all South American sequences, all mammalian sequences, and a representative subsample of North American avian sequences, including one representative virus from each cluster of viruses from the same US or Canadian state that were sampled from the same avian species on the same date. We used the Markov chain Monte Carlo (MCMC) method available in the BEAST package, v1.10.4[64], again using the NIH Biowulf Linux cluster, and a Bayesian non-parametric (SkyGrid) demographic model[65], with a GTR model of nucleotide substitution with gamma-distributed rate variation among sites. Each tip was assigned a location state and we performed a phylogeographic discrete trait analysis[66] to examine the routes of viral spatial dissemination between North America and South America, within South America, and within Peru. The MCMC chain was run separately four times for each dataset using the BEAGLE 3[67] library to improve computational performance, until all parameters reached convergence, as assessed visually using Tracer v.1.7.2[68]. At least 10% of the chain was removed as burn-in, and runs for the same dataset were combined using LogCombiner v1.10.4. A MCC tree was summarised using TreeAnnotator v.1.10.4 and visualised in FigTree v1.4.4. SPREAD 4[69] was used to visualise the spatial-temporal reconstruction. XMLs, MCC and ML trees, and GISAID acknowledgement tables are available in a GitHub repository (https://github.com/mostmarmot/Peru_AIV/).

### Reporting summary

Further information on research design is available in the Nature Portfolio Reporting Summary linked to this article.

## Data availability

We gratefully acknowledge the authors and both originating and submitting laboratories of the sequences from GISAID's EpiFlu™ Database on which this research is based. The GISAID server was unable to download a large table of 4000+ bird viruses used in our analyses, but tables for all South American and mammalian viruses used are provided in the supplemental materials. We also provide GenBank accession numbers with hyperlinks for all the sequences generated as part of this study in Supplementary Table 4. In addition, XMLs, MCC and ML trees, and GISAID acknowledgement tables are also available in a GitHub repository (https://github.com/mostmarmot/Peru_AIV/).

## Code availability

For NGS data analysis we used a suite of bioinformatic tools, including BBDuk and SPAdes, that are contained within the Geneious Prime 2023.0.4 package. In addition, XMLs, MCC and ML trees, and GISAID acknowledgement tables are available in a GitHub repository (https://github.com/mostmarmot/Peru_AIV/).

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

## Acknowledgements

Authors are employees of the Peruvian Ministerio de Desarrollo Agrario y Riego (MIDAGRI), of the Pontificia Universidad Catolica del Peru (PUCP), of the Wildlife Conservation Society, of the University of California - Davis, and of the US Government. This work was prepared as part of their official duties, with additional support from the National Institute of Allergy and Infectious Diseases at the National Institutes of Health under award U01AI151814 (M.L., A.G.G., B.M.S., D.J., P.B., C.K.J., C.C.M. and M.M.U.); from the Gordon and Betty Moore Foundation through Grant GBMF10809 to WCS Peru (P.C.C.); from the Intramural Research Program of the US National Library of Medicine at the NIH and the Centers of Excellence for Influenza Research and Response, National Institute of Allergy and Infectious Diseases, National Institutes of Health (NIH), Department of Health and Human Services, under contract 75N93021C00014 (M.I.N.). The funders had no role in study design, sample collection, data collection and analysis, decision to publish, or preparation of the article. The authors thank Sherilym Castillo, Esthefany Ruíz, Pedro Ramírez, Max Guerra, Antero Martínez, Wendy Rojas, Renato Colán, José Cerón, and Karina Espinoza from SERFOR for support during field work; Julio Reyes and Joe Macalupú from IMARPE for support with dolphin species identification and sampling permissions, respectively; Mariana Peralta from ATFFS-Arequipa, as well as Ferdinand Ortiz Mamani, Daniel Alama and Paul Cieza from SENASA for support with sample conservation and transport from field to lab; Miryam Quevedo from UNMSM for providing equipment for fieldwork; and Rosa Vento and Jorge Martínez from WCS for general logistical support.

## Author contributions

Study Design: M.L. and J.L. Field work and sample collection: J.J., W.S., K.P., L.A., P.C.C. and J.L. Lab work and sample processing: A.G.G., D.J., P.B. Bioinformatics processing and data analysis: M.L., A.G.G., B.M.S., D.J., P.B., M.I.N. Manuscript preparation: M.L., A.G.G., C.C.M., L.A., C.K.J., M.M.U., M.I.N. Manuscript editing: all authors. Review and approval of final manuscript: all authors.

## Competing interests

The authors declare no competing interests.
