## [Peer Review File · Nature Communications]

Highly pathogenic avian influenza A (H5N1) in marine mammals and seabirds in PeruREVIEWER COMMENTS

Reviewer #1 (Remarks to the Author):

Major points

1. Since wildlife in Peru is experiencing a mass die-off, pathological results, FISH, and TEM images from dead animals, particularly sea mammals, are of vital importance to characterize the biological properties of this novel reassortant.
2. Beside AIV, are there anything else found in the NGS data?
3. Figure 1A is of low resolution. Figure 1B is kind of weird, why LPAI was included? I suggested that the authors use the reference dataset provided by WHO/OIE/FAO.

Minor points

1. A/chicken/Ecuador/02/2022 should be highlighted in Figure 3B.
2. In Figure 4. Why the authors used A/H5N1 goose/Guangdong as a reference to identify the 70 variable sites, but used A/Vietnam/1203/2004 as a reference for visualization? This would cause potential confusion.
3. What do the 'X' symbols in Figure 4 mean? Figure 4 is huge and I suggest that those additional 30 sites without clear biological functions should be provided as a supplementary item.
4. Lines 200-203: please provide the detailed mutations found in the Peruvian strains related to the altered phenotypes.

Reviewer #2 (Remarks to the Author):

NCOMMS-23-10276-T Highly pathogenic avian influenza A (H5N1) in marine mammals and seabirds in Peru

In this manuscript Leguia and colleagues describe the sampling of a variety of animal species in Peru and the following detection and genetic characterization of HPAI H5N1 clade 2.4.3.3.b. This is the first scientific manuscript describing the genetic characterization of HPAI A/H5N1 in marine birds and mammals from South America.

The manuscript is well written and easy to read. It describes very important and timely data and all relevant information is provided.

Below a few minor comments to consider.

Line 48-49: please add reference.

Line 74-85: consider already mentioning the total number of samples obtained (now mentioned in the results at line 140)

Line 98-100: Please add minimal depth (number of reads) per region to fulfill QC for a particular region in the genome.

Line 160: I assume the PB2/PB1/Np and NS gene segments originate from LPAI strains in the 4:4 reassortment. Please confirm. Can the authors comment on the subtypes of these LPAI?

Line 157: not exactly clear why the authors use the phylogenetic tree of PA and PB2 in figure two instead of all 8.

Line 170-184: The authors conclude that there have been multiple HPAI H5N1 introductions and reassortments since the first introduction of 2.4.3.3b in North America. From the text it is not entirely clear if the authors can conclude (based on current available data) where the observed reassortment took place. Is this a novel 4:4 that likely reassorted in Peru or can they comment on origin regions of the other segments (and thereby location of reassortment) (the authors mention they cannot distinguish direction between Chile and Peru but to me it was not clear if they can say anything about where the reassorted virus originated from in general)

Line 191 and further: it is not entirely clear how the authors define "variable sites". Is this comparing 1) all obtained sequences or 2) bird vs mammal sequences or 3) comparing the Peruvian sequences with HPAI H5N1 from other regions. Especially the comparison between bird and mammal sequences that are time and location matched can provide evidence in routes of mammalian adaptation (either "known" substitutions or novel).

In addition the number of "variable sites" seem very high considering the close clustering of all sequences in the phylogenetic tree. How does this variation compare to other outbreaks from particular regions (so how much variation between sequences from outbreaks with multiple available sequences.) Can the authors comment on that?

Did the authors observe PB2 T271A which was observed in the mink outbreak in Spain?

Line 234-235: If any additional sequences are now available it would be extremely valuable to add those in the manuscript.

Figure 1: in the current colors and resolution and size of symbols of the figure it is hard to distinguish the symbols for negative and positive cases.

I hope the authors continue this important work and I encourage them to share their data and sequences in a timely manner e.g. via NCBI or GISAID.

Reviewer #3 (Remarks to the Author):

In this manuscript, Leguia et al. characterise eight HPAI H5N1 virus genomes from marine mammals and birds in South America. Maximum likelihood phylogenetic analysis of HA and NA gene segments confirms that these virus genomes belong to the HPAI HA lineage 2.3.4.4b. Phylogenetic analyses of the eight gene segments further indicate that Peruvian viruses belong to the reassortant R6 lineage, comprising 4 segments of the American clade C lineage and 4 segments of the Eurasian lineages. This lineage has become dominant in the Americas since the autumn of 2022. The authors found similar evidence for reassortment between American and Eurasian lineages for other H5N1 viruses detected in North and South America, suggesting that the Eurasian H5N1 lineage has reassorted with endemic lineages in North America on multiple occasions since 2021. Lastly, by comparing these virus genomes with A/H5N1/Goose/Guangdong/1996 virus reference sequence, they identified 70 variable sites, 30 of which have not been previously reported or characterised.

The study is well written and makes a valuable contribution by reporting the first HPAI H5N1 virus genomes from marine mammals and birds in South America, a genomically undersampled region for AIV. However, I do believe that the analyses currently presented are quite limited and could be further expanded to strengthen the study findings. For example, undertake a phylodynamic analysis to date the arrival of the R6 lineage into Chile and Peru and estimate the evolutionary rates to assess if it is evolving unusually faster or as expected. I also have some questions about their current analysis, which I have detailed below. I

Major comments:

- 1) The phylogeny in Figure 1 is problematic because you have LPAI viruses intermingled with HPAI lineages, when the HPAI lineages should be more closely related to each other. It does seem strange to use lineage classification based on HA but infer the phylogeny on concatenated HA and NA alignments (where NA belongs to different subtypes). To avoid generating confusion and misinterpretation, it would be better to show the HA and NA phylogenies separately or just present the HA phylogeny.
- 2) The claim that R6 viruses first came to Peru and then spread to Chile cannot be discerned based on the phylogenetic analysis presented in Figure 3B alone, as the branches leading to the Chilean and Peruvian viruses are polytomies. Is the inference of the authors based on other information, in addition to phylogeny, such as the migratory patterns of wild birds? If so, they should clarify these details in the main text. However, from the current phylogenetic analysis, we cannot distinguish if the viruses arrived in Peru and Chile at the same time, Peru first and then Chile, or Chile first and then Peru. Given the limited genomic sampling in South America, the authors must be cautious when interpreting their findings.
- 3) It is not clear if the phylogenetic analysis in Figure 3B is also based on concatenated HA and NA? If whole genome sequences are available for most sequences, it might be worth inferring a phylogeny based on the concatenated gene segments to improve phylogenetic resolution (e.g., might help better resolve the branching order in the Peruvian outbreak). Concatenating gene

segments here should not be problematic (in contrast to point 1) when examining within-lineage divergence.

4) The authors report 30 variable sites that are limited to viruses from Latin America and suggest that this indicates that 'the virus is indeed changing as it travels south from the northern hemisphere). Given that it is a fast-evolving RNA virus, mutational changes in the virus genome are expected. Therefore, observation of 30 variable sites is an especially notable result. Molecular clock analysis would help determine whether South American viruses are evolving particularly faster compared to North American or Eurasian isolates. Breaking down the evolutionary rate into non-synonymous and synonymous rates would be valuable for assessing differences in selective pressures (e.g. renaissance counting in BEAST v1).

Minor comments:

1) Line 170 – Page 4: I think March 2021 should be March 2022 based on Figure 3A and Table 2

2) Are sequences available from poultry outbreaks during the same time period in Peru? If so, how do these relate to viruses from marine mammals and birds?

H5N1 REVIEWER COMMENTS

Reviewer #1 (Remarks to the Author):

Major points

1. Since wildlife in Peru is experiencing a mass die-off, pathological results, FISH, and TEM images from dead animals, particularly sea mammals, are of vital importance to characterize the biological properties of this novel reassortant.

- We agree with the reviewer that these types of data are important in outbreak situations like the one described in our paper. However, we do not currently have pathology results, nor FISH or TEM data, because of the context in which we carry out our work in-country. In the best of circumstances, such data are difficult to generate in Peru primarily because there is little or no access to the instruments and/or reagents needed for FISH or TEM, particularly in an outbreak situation. In the case of the pathology reports, there is only one such report (done on the aborted sea lion) that concluded that all tissues looked “normal” even though no images were taken or attached to the report. The Genomics Laboratory at PUCP, which carried out all the molecular biology and genomics work described in this paper, works in collaboration with ministry of agriculture/health employees who are responsible for sample collection. In that sense, we can make recommendations regarding how samples should be collected, handled, and further analyzed, but unfortunately our recommendations are non-binding, which often leads to “gaps” in the analysis. This is something we are actively trying to change.

2. Beside AIV, are there anything else found in the NGS data?

- Not from the various analyses that we carried out. All samples were tested for coronaviruses, flaviviruses, alphaviruses and bunyaviruses using pan-PCR assays. All samples were negative for these viral family tests. We now include a sentence (line 94 in the tracked changes version) that reads “All samples were also tested for coronaviruses, alphaviruses, bunyaviruses and flaviviruses using pan-PCR assays, and all samples were negative for these tests (not shown).” Please note that the genomic sequence data was generated using targeted approaches that selectively amplify influenza genomes, so we were not able to “mine” those NGS data for pathogen discovery. In the few cases where we also applied shotgun sequencing approaches that support pathogen discovery, we did not see evidence of any other pathogen that could potentially explain the observed disease etiologies.

3. Figure 1A is of low resolution. Figure 1B is kind of weird, why LPAI was included? I suggested that the authors use the reference dataset provided by WHO/OIE/FAO.

- Figure 1 has been re-worked for several reasons, including the fact that other reviewers also commented on the fact that the phylogeny in 1B could be confusing. Below we address the specific comments of this reviewer regarding Figure 1.
- We now provide a new, high-resolution map for Figure 1A.

- In the original phylogeny listed in Figure 1B, we used LPAI as an “outgroup” given that the figure was intended as a “big picture” view of what was going on with Latin American strains. However, since the figure was confusing, we have removed it and replaced it with field images of the animals tested. Instead, we now rely solely on the robust and complete phylogenies provided in Figures 2 and 3.
- In the original phylogeny of Figure 1B we included some references from the dataset provided by WHO/OIE/FAO, particularly for outgroups, but since that dataset is from 2014 (essentially 9 yrs. old), it does not contain 2.3.44 strains, which are more recent and more appropriate for what we wanted to show, which was a “big picture” view of what was circulating in the region. However, since Figure 1B has been confusing, we have now removed it and rely solely on the phylogenies provided in Figures 2 and 3.

Minor points

1. A/chicken/Ecuador/02/2022 should be highlighted in Figure 3B.

- A/chicken/Ecuador/02/2022 and A/chicken/Ecuador/03/2022 are both highlighted in the revised Figure 3A. In addition, we now cite the following reference which discusses H5N1 in Ecuador and which became available while our manuscript was under review: <https://www.sciencedirect.com/science/article/pii/S1201971223005337>

2. In Figure 4. Why the authors used A/H5N1 goose/Guangdong as a reference to identify the 70 variable sites, but used A/Vietnam/1203/2004 as a reference for visualization? This would cause potential confusion.

- Yes, we now see how this can be confusing. Both references were used in the original version of the manuscript for two reasons: 1) because A/H5N1 goose/Guangdong is the first ever H5N1 described and 2) because A/Vietnam/1203/2004 is the reference used by the CDC table. However, to avoid confusion, and because there are differences between the Guangdong and Vietnam references, we now redefine what we consider a “variable site” as a site that is different from both the Guangdong and Vietnam references. This is explicitly mentioned in the revised manuscript (line 180 in tracked changes version) and reads: “We focused on variable sites that were different from both the original A/H5N1 goose/Guangdong strain from 1996³⁹ and the A/Vietnam/1203/2004 strain used to annotate amino acid positions in the CDC inventory³².”
- In addition, we introduce an even tighter definition of a “variable site” by including the following text (line 192 in tracked changes version): “However, given the increased age of the two reference sequences used to define variable sites, which would make them unsuitable to identify potentially relevant recent changes, we also included in the analysis two additional references representing R6 reassortants from 2022 identified in birds (A/chicken/Wyoming/22-032216-001/2022) and mammals (A/skunk/Washington/22-019274-001/2022).” The result of this change is a smaller number of sites listed, each with higher importance.

3. What do the 'X' symbols in Figure 4 mean? Figure 4 is huge and I suggest that those additional 30 sites without clear biological functions should be provided as a supplementary item.

- Figure 4 is now significantly pared down based on a new, more useful definition of what constitutes a “variable site” and, equally important, on new sequence data that became available since our original submission and that allowed us to eliminate sites that had been previously listed as potentially interesting.
- “Xs” mean Ns, or gaps in sequence. We opted not to list them as Ns since we are displaying amino acids rather than nucleotides, and in amino acid nomenclature N = asparagine.

4. Lines 200-203: please provide the detailed mutations found in the Peruvian strains related to the altered phenotypes.

- We are unsure what the reviewer is referring to. The mutations are fully listed and provided for all data generated as part of this work, both in Figure 4 and in Supplemental Table 2.

Reviewer #2 (Remarks to the Author):

NCOMMS-23-10276-T Highly pathogenic avian influenza A (H5N1) in marine mammals and seabirds in Peru

In this manuscript Leguia and colleagues describe the sampling of a variety of animal species in Peru and the following detection and genetic characterization of HPAI H5N1 clade 2.4.3.3.b. This is the first scientific manuscript describing the genetic characterization of HPAI A/H5N1 in marine birds and mammals from South America. The manuscript is well written and easy to read. It describes very important and timely data and all relevant information is provided.

Below a few minor comments to consider.

Line 48-49: please add reference.

- We thank the reviewer for noticing this omission on our part. New references have been added.

Line 74-85: consider already mentioning the total number of samples obtained (now mentioned in the results at line 140)

- The total number of samples obtained (69) is now mentioned earlier (line 80 in tracked changes version). We have tested all the samples obtained. In essence, samples obtained = samples tested.

Line 98-100: Please add minimal depth (number of reads) per region to fulfill QC for a particular region in the genome.

- We thank the reviewer for noticing this omission on our part. These data are now fully listed in Supplement Table 1 for all genomic regions with 100% length of coverage.

Line 157: not exactly clear why the authors use the phylogenetic tree of PA and PB2 in figure two instead of all 8.

- The phylogenetic trees of PA and PB2 were representative of the two different types of topologies we were obtaining depending on which genomic segment was being used to generate the trees. Because the trees were so large, we initially opted for showing only two that were representative of all segments. However, we have found a way to display all trees in one figure, so to avoid confusion and enable direct comparison of all tree topologies, Figure 2 now shows all 8 trees, as suggested.

Line 160: I assume the PB2/PB1/Np and NS gene segments originate from LPAI strains in the 4:4 reassortment. Please confirm. Can the authors comment on the subtypes of these LPAI?

- As mentioned above, Figure 2 now shows all 8 trees, as suggested. In addition, the trees are “global H5N1 phylogenies,” meaning they have been constructed using all AIVs collected globally and submitted to GISAID since January 1, 2021. In order to provide additional information and clarity on the data presented, we now include a note (line 513 in tracked changes version) explaining that “raw tree files for each genome segment are available at GitHub (https://github.com/mostmarmot/Peru_AIV/).”

Line 170-184: The authors conclude that there have been multiple HPAI H5N1 introductions and reassortments since the first introduction of 2.4.3.3b in North America. From the text it is not entirely clear if the authors can conclude (based on current available data) where the observed reassortment took place. Is this a novel 4:4 that likely reassorted in Peru or can they comment on origin regions of the other segments (and thereby location of reassortment) (the author mention they cannot distinct direction between Chili and Peru but to me it was not clear if they can say anything about where the reassorted virus originated from in general)

- We apologize for the confusion here. The 4:4 reassortment event was detected in North America prior to the detection of these reassortant viruses in Peru. The Results section “Eurasian-American lineage reassortants” (line 106 in the tracked changes version) and Figure 3 now describe the emergence and spread of the 4:4 “R6” reassortants in detail, including a calculation of first introduction of H5N1 into Peru/Chile based on MCC trees generated using Bayesian approaches.

Line 191 and further: it is not entirely clear how the authors define “variable sites”. Is this comparing 1) all obtained sequences or 2) bird vs mammal sequences or 3) comparing the Peruvian sequences with HPAI H5N1 from other regions. Especially the comparison between bird and mammal sequences that are time and location matched can provide evidence in routes of mammalian adaptation (either “known” substitutions or novel).

- We thank the reviewer for this observation, which echoes concerns voiced by Reviewer #1. We acknowledge that part of the confusion stems from the fact that in some cases we referred to the “variable sites” in relationship to changes with respect to the original Guangdong reference, whereas in others we referred to them in relationship to the CDC inventory, which doesn’t use Guangdong as a baseline. This is now corrected as we have explicitly narrowed down the definition of what constitutes a “variable site” (lines 180

and 194 in the tracked changes version) and, based on this new, tighter definition, we have updated Figure 4.

- “Variable sites” are now defined as “sites that were different from both the original A/H5N1 goose/Guangdong strain from 1996³⁹ and the A/Vietnam/1203/2004 strain used to annotate amino acid positions in the CDC inventory³²” (line 180 in the tracked changes version) and also different from “two additional references representing R6 reassortants from 2022 identified in birds (A/chicken/Wyoming/22-032216-001/2022) and mammals (A/skunk/Washington/22-019274-001/2022)” (line 194 in the tracked changes version).
- With this new definition we are able to identify mutations that cluster in birds vs. mammals, as indicated both in Supplement Table 2 and Figure 4.

In addition, the number of “variable sites” seem very high considering the close clustering of all sequences in the phylogenetic tree. How does this variation compare to other outbreaks from particular regions (so how much variation between sequences from outbreaks with multiple available sequences.) Can the authors comment on that?

- We agree that in the original submission, the number of variable sites was high. In part that was because originally, we had less sequence data than we do now, as new genomic data was generated while the paper was in peer review. Based on this additional data, and on the new more stringent definition of “variable sites”, we now have a much-reduced number of mutations that are of interest because they repeat in multiple samples and/or cluster either in mammals or in birds. This reduced number of variable sites of interest is fully displayed in Figure 4.

Did the authors observe PB2 T271A which was observed in the mink outbreak in Spain?

- No. Our sequences all have T in position 271.

Line 234-235: If any additional sequences are now available it would be extremely valuable to add those in the manuscript.

- Additional samples have become available since our initial submission and those have been incorporated into the analysis, text and figures throughout the manuscript.

Figure 1: in the current colors and resolution and size of symbols of the figure it is hard to distinguish the symbols for negative and positive cases.

- Figure 1 is now completely re-worked and includes a high-resolution map.

I hope the authors continue this important work and I encourage them to share their data and sequences in a timely manner e.g. via NCBI or GISAID.

- Thank you and yes, all data has been deposited in public repositories.

Reviewer #3 (Remarks to the Author):

In this manuscript, Leguia et al. characterise eight HPAI H5N1 virus genomes from marine mammals and

birds in South America. Maximum likelihood phylogenetic analysis of HA and NA gene segments confirms that these virus genomes belong to the HPAI HA lineage 2.3.4.4b. Phylogenetic analyses of the eight gene segments further indicate that Peruvian viruses belong to the reassortant R6 lineage, comprising 4 segments of the American clade C lineage and 4 segments of the Eurasian lineages. This lineage has become dominant in the Americas since the autumn of 2022. The authors found similar evidence for reassortment between American and Eurasian lineages for other H5N1 viruses detected in North and South America, suggesting that the Eurasian H5N1 lineage has reassorted with endemic lineages in North America on multiple occasions since 2021. Lastly, by comparing these virus genomes with A/H5N1/Goose/Guangdong/1996 virus reference sequence, they identified 70 variable sites, 30 of which have not been previously reported or characterised.

The study is well written and makes a valuable contribution by reporting the first HPAI H5N1 virus genomes from marine mammals and birds in South America, a genomically undersampled region for AIV. However, I do believe that the analyses currently presented are quite limited and could be further expanded to strengthen the study findings. For example, undertake a phylodynamic analysis to date the arrival of the R6 lineage into Chile and Peru and estimate the evolutionary rates to assess if it is evolving unusually faster or as expected. I also have some questions about their current analysis, which I have detailed below. I

Major comments:

1) The phylogeny in Figure 1 is problematic because you have LPAI viruses intermingled with HPAI lineages, when the HPAI lineages should be more closely related to each other. It does seem strange to use lineage classification based on HA but infer the phylogeny on concatenated HA and NA alignments (where NA belongs to different subtypes). To avoid generating confusion and misinterpretation, it would be better to show the HA and NA phylogenies separately or just present the HA phylogeny.

- We thank this reviewer for critical analysis and helpful suggestions on our figure, and we comment that all three reviewers have agreed that this figure was confusing. As a result, we have eliminated the phylogeny in Figure 1 and we now rely solely on single gene phylogenies of all genomic segments, as shown in Figure 2.

2) The claim that R6 viruses first came to Peru and then spread to Chile cannot be discerned based on the phylogenetic analysis presented in Figure 3B alone, as the branches leading to the Chilean and Peruvian viruses are polytomies. Is the inference of the authors based on other information, in addition to phylogeny, such as the migratory patterns of wild birds? If so, they should clarify these details in the main text. However, from the current phylogenetic analysis, we cannot distinguish if the viruses arrived in Peru and Chile at the same time, Peru first and then Chile, or Chile first and then Peru. Given the limited genomic sampling in South America, the authors must be cautious when interpreting their findings.

- This is a good point. Originally, we presumed the R6 viruses arrived in Peru before Chile based on geography and the Pacific flyway, not the phylogeny, as Peru is north of Chile and wild birds migrating south from North America would pass through Peru before arriving in Chile. But, as the reviewer points out, the phylogenetic tree cannot discern which country the viruses arrived at first, and bird migration patterns are complex. We

have revised the text accordingly (line 142 onwards in the tracked changes version) and performed a new phylogeographic analysis using BEAST (Figure 3A) to show quantitatively that the posterior probabilities for the location at this node cannot distinguish whether Peru or Chile received R6 viruses first.

3) It is not clear if the phylogenetic analysis in Figure 3B is also based on concatenated HA and NA? If whole genome sequences are available for most sequences, it might be worth inferring a phylogeny based on the concatenated gene segments to improve phylogenetic resolution (e.g., might help better resolve the branching order in the Peruvian outbreak). Concatenating gene segments here should not be problematic (in contrast to point 1) when examining within-lineage divergence.

- The reviewer is correct, it would make sense to concatenate the genomes for the phylogenetic analysis shown. Unfortunately, some key viruses from Peru, Chile, and other South American locations are missing one or more genome segments. If we removed each virus that did not have a complete genome, we would reduce the size of our dataset and lose important data. We therefore analyze each genome segment individually. Possibly in the future, when a larger number of viruses are available from Peru and Chile, it will be possible to look at transmission and spatial movements within Peru and Chile with concatenated genomes.
- We now explicitly state that the tree is based on the PB2 segment (line 507 in tracked changes version), and further, we provide all the raw data for the MCC tree through GitHub: https://github.com/mostmarmot/Peru_AIV/ (line 514 in tracked changes version).

4) The authors report 30 variable sites that are limited to viruses from Latin America and suggest that this indicates that 'the virus is indeed changing as it travels south from the northern hemisphere). Given that it is a fast-evolving RNA virus, mutational changes in the virus genome are expected. Therefore, observation of 30 variable sites is an especially notable result. Molecular clock analysis would help determine whether South American viruses are evolving particularly faster compared to North American or Eurasian isolates. Breaking down the evolutionary rate into non-synonymous and synonymous rates would be valuable for assessing differences in selective pressures (e.g., renaissance counting in BEAST v1).

- We agree that a molecular clock analysis would be helpful in general. However, given the new, tighter definition of what constitutes a “variable site” we have a lot fewer sites that would warrant further examination. Hopefully in the future, when more sequence data becomes available for these regional viruses, we will be able to complete such analysis in a meaningful way.

Minor comments:

1) Line 170 – Page 4: I think March 2021 should be March 2022 based on Figure 3A and Table 2

- Yes, March 2021 is a mistake, and it has now been corrected. Thank you for noticing that.

2) Are sequences available from poultry outbreaks during the same time period in Peru? If so, how do these relate to viruses from marine mammals and birds?

- Not from Peru. We initially thought there were sequences from chickens in Ecuador and Colombia, and at least 6 sequences from vultures in Brazil, but these turned out to be mislabeled and there are no Brazilian sequences to analyze.
- There was no genetic linkage between the viruses circulating in wild birds and marine mammals in Peru and the poultry viruses in Ecuador and Colombia, which were introduced separately from North America (Figure 3).

REVIEWERS' COMMENTS

Reviewer #2 (Remarks to the Author):

The authors nicely addressed all concerns and suggestions raised by the three reviewers. They explain in detail the changes and additions made and this has improved the content and clarity of the manuscript.

It is of particular appreciation that the authors have made the manuscript more timely by adding new sequences that have become available in the meantime, strengthening the value of the study.

I do not have any further questions or concerns.

Reviewer #3 (Remarks to the Author):

All my previous concerns have now been addressed. Many thanks to the authors for undertaking additional phylodynamic analyses, I think it helps strengthen the study by clarifying what we currently know about the introduction of 2.3.4.4.b/HPAI into South America. Overall, the study is well written and makes an important contribution to the ongoing HPAI outbreaks around the world.

I only have one minor comment. "Dessert" on page 6, line 28 should be "desert"